# Unveiling Health Inequalities: Exploring Metabolic Dysfunction in Rural Roma Communities

**DOI:** 10.3390/healthcare12080816

**Published:** 2024-04-11

**Authors:** Dana Crișan, Lucreția Avram, Cristiana Grapă, Andrada Nemeș, Maria-Virginia Coman, Mihail Simion Beldean-Galea, Radu-Tudor Coman, Tudor Călinici, Valer Donca, Rareș Crăciun

**Affiliations:** 1Faculty of Medicine, University of Medicine and Pharmacy “Iuliu Hatieganu”, 400000 Cluj-Napoca, Romania; crisan.dc@gmail.com (D.C.); nemes.andrada.raluca@elearn.umfcluj.ro (A.N.); coman.radu@umfcluj.ro (R.-T.C.); tcalinici@umfcluj.ro (T.C.); valerdonca@gmail.com (V.D.); craciun.rares.calin@elearn.umfcluj.ro (R.C.); 2Clinical Municipal Hospital Cluj-Napoca, 400139 Cluj-Napoca, Romania; 3“Prof. Dr. O. Fodor” Regional Institute of Gastroenterology and Hepatology, 400162 Cluj-Napoca, Romania; 4“Raluca Ripan” Institute for Research in Chemistry, “Babeş-Bolyai” University, 400294 Cluj-Napoca, Romania; virginia.coman@ubbcluj.ro; 5Faculty of Environmental Science and Engineering, “Babeş-Bolyai” University, 400294 Cluj-Napoca, Romania; simion.beldean@ubbcluj.ro

**Keywords:** Roma minority, metabolic syndrome, liver steatosis, healthcare

## Abstract

Background: Europe’s largest ethnic minority, the Roma, are often confronted with substantial obstacles that result in health disparities. Research indicates that there are elevated rates of both communicable and non-communicable diseases, such as metabolic syndrome (MetS), among Roma communities, often linked to living conditions, limited education, or poverty. This study centers on remote rural Roma settlements in Romania, evaluating the prevalence of metabolic dysfunction, obesity, and liver steatosis while considering socio-economic and lifestyle factors. Methods: Over a period of 36 months, local visits to a total of 25 rural Roma communities were conducted, where a medical team gathered information through a standardized questionnaire and conducted a physical exam on every participant. Liver steatosis was also recorded with the help of a portable wireless ultrasound device. Results: Our study included 343 participants, with a predominance of female subjects, representing 72.5% (*n* = 249) of the patients. The prevalence of obesity, defined by a body mass index (BMI) above 30 kg/m^2^, was 32.2% (*n* = 111). Arterial hypertension was found to have a prevalence of 54.1% (*n* = 185), with de novo hypertension being observed in 19.2% patients (*n* = 66). Type 2 diabetes mellitus was found in 28.9% patients (*n* = 99), with 19.5% being de novo cases. The prevalence of hepatic steatosis was 57.2% (*n* = 111/194). A positive association between metabolic features and at-risk behaviors was found. Conclusions: This study underscores the transition from infectious to metabolic diseases in vulnerable communities and highlights the urgency of targeted public health strategies tailored to the unique needs of rural Roma populations, aiming to mitigate health disparities and promote equitable healthcare access.

## 1. Introduction

The Roma constitute the largest ethnic minority in Europe, estimated to number between 12 and 15 million. They currently reside predominantly in Bulgaria, Romania, Slovakia, Hungary, the Czech Republic, and Slovenia, but migrations in the last two decades have led to a rise in numbers settling across European Union (EU) states [1,2]. Typically living in economically marginalized regions, they often inhabit segregated colonies marked by harsh living conditions. Regardless of location, common challenges faced by the Roma include poverty, limited educational opportunities, high unemployment rates, and social marginalization [3].

Given their predominantly low socio-economic status and its correlation with health outcomes, it is reasonable to infer that Roma are subject to poorer health and shorter life expectancies compared to the majority population, a notion frequently referenced in both research and public discourse [4,5]. However, due to a reluctancy to record the Roma ethnicity in some official documents, like medical records, and significant obstacles hindering data collection on minorities, such assumptions cannot be conclusively proven correct [6].

The prevalence of both communicable and non-communicable diseases is notably higher in Roma communities [7]. Conditions such as tuberculosis, hepatitis, and skin diseases are reported to be more widespread within Roma communities, largely due to their disadvantaged living conditions and inadequate hygiene practices [8]. However, there are fewer studies that assess other lifestyle-related diseases suffered by Roma subjects.

The incidence of metabolic syndrome (MetS) serves as a crucial measure for assessing health status and associated risks within a population. MetS encompasses a cluster of interconnected factors that significantly elevate the likelihood of developing cardiovascular diseases (CVDs), type 2 diabetes mellitus (T2DM), cancer, steatotic liver disease, dementia, infertility, and various other conditions [9]. According to the International Diabetes Federation, metabolic syndrome is diagnosed when an individual exhibits abdominal obesity alongside at least two of the following conditions: elevated blood pressure (BP) or ongoing treatment for hypertension, increased fasting plasma glucose (FPG) levels or a previous diagnosis of diabetes mellitus, high triglyceride (TG) levels or ongoing treatment for a lipid disorder, and decreased levels of high-density lipoprotein cholesterol (HDL-C) or ongoing treatment for a lipid disorder [10].

Studies on the prevalence and characteristics of MetS in Roma populations have identified an increasing prevalence of hypertension and central obesity, with one study pointing out that these changes are found predominantly in the female population in the 20–34 age group and in both sexes in the 35–49 age group [11]. A study conducted by Kósa et al. revealed a significant association between metabolic syndrome and the deteriorating health status observed in the Roma community: notably, decreased levels of HDL cholesterol and elevated fasting blood glucose, particularly when influenced by genetic factors, emerged as prominent factors in metabolic syndrome [12]. Another study conducted on 452 Roma subjects living in segregated communities called attention to the increased prevalence of MetS, hepatitis B and E, and other parasitic diseases, underscoring the necessity of regular screening and targeted interventions for reducing health inequalities [13]. Additionally, a recent meta-analysis [14] signaled an increased prevalence of diabetes among Romani individuals compared to Caucasians; however, none of these studies met the criteria for representative samples and an adequate number of cases, thus precluding definitive conclusions. The cited study further highlights the need for additional research on this subject.

Our study aimed to assess the primary health-related concerns in isolated rural Roma communities in North-Western and Central Romania, focusing on obesity, metabolic dysfunction, and steatotic liver disease. As a secondary aim, we intended to identify the most prevalent metabolic ailments and correlate them with socio-economic conditions, unhealthy lifestyle choices, and obstacles hindering healthcare access in this demographic. The rationale for conducting this study was to increase awareness of the substantial disparities in socio-economic development and health indicators between Roma and non-Roma populations across Europe, focusing specifically on vulnerable rural communities in Romania, providing valuable insights into the prevalence of metabolic diseases and associated risk factors within this context.

## 2. Materials and Methods

### 2.1. Study Design and Workflow

A total of 25 vulnerable rural Roma communities in North-Western and Central Romania were selected following a preliminary assessment conducted in collaboration with a team comprising minority representatives, mediators, local primary care providers, and public health experts from the National Institute of Public Health, underscoring the collaborative nature of our research. After community selection, local visits were conducted by a team of five medical experts in internal medicine, gastroenterology, and hepatology. The initial timeframe for the project development was 18 months. However, the timeframe was extended to 36 months due to restrictions imposed because of the COVID-19 pandemic.

A standardized health-related quality-of-life questionnaire was developed by the medical team (see Appendix A); it included demographic variables, anthropometric variables, medical history, health-related behaviors, risk factors (including smoking, dietary patterns, sweetened beverage and alcohol consumption, and physical activity), and socio-economic indicators (access to primary care, sanitation, and workforce involvement). At-risk consumption of sweetened beverages (containing in excess of 35 kCal or 9 g of sugar/100 mL) was set at 500 mL/day. The questionnaire was subsequently adapted to fit the social, cultural, and educational profiles of the respondents in collaboration with community representatives and local primary care providers. The questionnaire was applied on site by the team of medical experts, and the results were recorded in a case report form (CRF). Given the vulnerable status of Roma communities, stringent measures were employed to protect participants’ identities and ensure their autonomy in decision making. Informed consent was obtained through transparent communication, with participants fully understanding the study’s purpose and potential risks. Participation was voluntary, with individuals being able to withdraw at any stage without consequence. Anonymity was preserved in data collection and reporting to prevent stigmatization or discrimination.

A general physical exam assessing cardiovascular, respiratory, and digestive systems was conducted. We recorded anthropometric variables (abdominal circumference, weight, and height), blood pressure, and blood sugar levels using portable blood sugar monitors. Stool samples were collected for parasitological exams and analyzed in a tertiary care laboratory using the current standard of care. Screening for hepatic steatosis and features of advanced liver disease was performed using a portable wireless ultrasound (US) probe (C10RN Wireless Black & White Abdominal Ultrasound Convex Probe, Konted Medical Technology Co., Ltd., Beijing, China). Hepatic steatosis was visually assessed by a certified EFSUMB Level 3 ultrasonographer, using features such as acoustic attenuation, overall parenchymal brightness, and gross hepato-renal index. Features of advanced liver disease, such as liver morphology nodularity, portal vein enlargement, collateral circulation, and spleen size, were also screened.

### 2.2. Statistical Analysis

Categorical data were presented as counts and percentages. Comparisons of categorical data were performed with the Chi-square test or Fisher’s exact test in case of low expected frequencies. Continuous normally distributed data were presented as means and standard deviations, while continuous variables with a non-normal distribution were expressed as medians and inter-quartile ranges (IQR). The comparison was performed using the Mann–Whitney U test. The threshold for statistical significance was set at *p* = 0.05. Our analysis focused on determining if metabolic features (diabetes mellitus or arterial hypertension) were associated with at risk-habits in the targeted population; we also tested associations between different demographic features and at-risk habits and liver steatosis. The study size was determined based on the anticipated prevalence of health-related concerns within the targeted Roma communities and the available resources for data collection and analysis. Quantitative variables were handled according to established statistical methods, with appropriate groupings chosen based on the nature of the data and analytical requirements. All statistical methods, including those used for controlling confounding and examining subgroups and interactions, were clearly described. Sensitivity analyses were conducted to assess the robustness of the findings. SPSS software version 29.0.1.0 (SPSS Inc., Chicago, IL, USA) was used for the statistical analysis, which was designed and performed by a certified biomedical statistician.

## 3. Results

In this study, we analyzed data from 343 participants, representing 34.3% percent of the target population, initially calculated to be 1000 participants. The target population was below expectations, as visits were hindered by some obstacles: field trips had been canceled or postponed due to quarantine and other regulations imposed to combat the COVID-19 pandemic. Furthermore, the target population was reluctant to participate in medical activities, citing cultural beliefs. Poor enrollment of male participants was observed due to informal work-related errands.

There was a significant predominance of female subjects, representing 72.5% (*n* = 249) of the patients. The median age was 51 years old (IQR 38.25–63). Risk factors such as smoking and alcohol consumption were found for 46.4% (*n* = 159) and 11.1% (*n* = 38) of the Roma participants, respectively. Notably, the prevalence of smoking was more than threefold greater than that of alcohol consumption. For this study, any consumption of alcohol reported was considered an at-risk behavior and included in the statistical analysis. More than half of the population studied (51.2%, *n* = 175) reported an at-risk consumption of sweetened beverages. Regarding healthcare access, most subjects (*n* = 324, 94.5%) reported having formal access to primary care such as a family doctor or a general practitioner. However, fewer than half of the participants were compliant with routine (yearly or more frequent) visits. A small percentage had stable employment (*n* = 33, 9.6%), with the majority reporting that they either worked part-time, had informal jobs, or had never had a job.

Concerning weight-related issues, the prevalence of a body mass index (BMI) exceeding 25 kg/m^2^ was 67.9% (*n* = 233), with 32.3% of the participants (*n* = 111) having various degrees of obesity. The prevalence of arterial hypertension was 54.1% (*n* = 185), and a significant rate of de novo hypertension was observed, namely, 19.2% (*n* = 66). The overall prevalence of type 2 diabetes mellitus was 28.9% (*n* = 99), with 19.5% of the cases being diagnosed during medical visits. The prevalence of hepatic steatosis was 57.2% (*n* = 111/194) in the screened population. No patients with features of advanced liver disease were identified. The prevalence of a positive parasitological exam was 5.8% (*n* = 7/120), with all cases being positive for *Giardia lamblia*. The baseline characteristics of the study population, health-related risk factors, and disease burden are depicted in Table 1.

In the context of the high prevalence of metabolic features in this study group, we tested the positive association between different metabolic features and at-risk behaviors (Table 2). A significant association was identified between DM and HTN (*p* < 0.001), as expected, as both conditions share common risk factors such as obesity and insulin resistance. We then proceeded to test if at-risk sweet-beverage consumption is associated with metabolic features and found a positive association with DM (*p* = 0.001) and newly diagnosed HTN (*p* = 0.004). Regarding another at-risk behavior, alcohol consumption, our analysis revealed it was correlated with de novo HTN (*p* = 0.004) but not with DMA.

To assess the factors associated with hepatic steatosis for this study group, we analyzed different habits and metabolic and demographic features (Table 2). The presence of liver steatosis in the Roma population was notably elevated in the obesity group (*p* < 0.001) and among individuals with DM (*p* = 0.0007) and hypertension (*p* < 0.001). However, no significant association was found between liver steatosis and alcohol consumption.

## 4. Discussion

Our results revealed a disproportionately high burden of metabolic risk factors in the Romanian rural Roma communities compared to the general population, implying that amendable lifestyle-related risk factors constitute the most pressing healthcare issue in these groups. Deeply intertwined, the prevalence of overweight and morbid obesity, T2DM, arterial hypertension, and MASLD far exceed other conditions as the primary morbid conditions.

For an adequate understanding of the upcoming discussion, a brief overview of the socio-economic and cultural characteristics of rural Roma communities is warranted. Facing significant challenges, these communities often grapple with poverty, limited access to education, and healthcare disparities. Economic marginalization is evident, with many relying on informal and seasonal labor, facing high unemployment rates, and struggling with substandard living conditions. Education gaps persist, hindering social mobility and contributing to a cycle of disadvantages.

First, regarding weight status, our results are in line with previous reports on other rural Roma communities, in which a 70.5% prevalence of a BMI exceeding 25 kg/m^2^, and 45.2% for obesity, was reported [15]. In contrast, the prevalence of overweight and obese patients in the general population was significantly lower, at 31.1% and 21.3%, respectively [16], figures which resemble the European Commission estimates for 2019, predicting that 52.7% of adults residing in the European Union will be overweight (as reported and accessed on 4 February 2024). The disparity between vulnerable communities and the general population regarding their weight statuses has been recently studied in a multicentric study that included approximately 25,000 participants from low-socio-economic-status communities from six European countries. The authors of the cited study created a three-variable score comprising unemployment, financial instability, and low education level (<12 years of formalized education), factors that were directly proportional to the probability of being overweight or obese, with overall rates of 34.5% and 15.8%, respectively [17]. The higher figures for both overweight and morbidly obese status in our study are plausibly related to the Romani’s proximity to the more severe end of the socio-economic and cultural impoverishment scale among European communities [18,19,20].

Insulin resistance and T2DM are in close correlation with and occur as a direct pathophysiological consequence of being overweight and obese. According to a large-scale observational study, the age-standardized prevalence rate for T2DM ranges between 9% and 11% in Central and Western European countries [21], while a recent Romanian report estimated the prevalence of T2DM at 11.6% among a randomly selected population screened in a primary care setting, of which up to one quarter of cases were previously undiagnosed [22]. The figures from our study are substantially higher, with an overall prevalence of 28.9%. There appear to be significant disparities in the prevalence of diabetes among ethnic minorities residing in Europe. According to a systematic review and meta-analysis published in 2015, the odds ratio for different subsets of ethnic minorities for T2DM ranged from 1.3 to 3.7 [23]. While the impact of a different genetic background undoubtedly accounts for variations in prevalence, the common characteristic for most of these communities is a lower socio-economic status compared to the local population. Much like weight-related issues are associated with a lower socio-economic status, evidence suggests that substandard health-related behaviors, limited access to healthcare, and employment are associated with an increased risk of developing diabetes, and these issues appear to more frequently affect ethnic minority groups [24].

Another concerning finding of our study was the extremely high prevalence of hepatic steatosis in the absence of ultrasonographic findings of advanced liver disease. The prevalence of non-alcoholic fatty liver disease in the general population was estimated at 24% [25], and in the epidemiologic context of this discussion, it appears safe to assume that the recent nomenclature change to Metabolic-Dysfunction-Associated Steatotic Liver Disease (MASLD) does not significantly alter the value of prior estimates [26]. Consequently, our results represent an almost-threefold uptick in the prevalence of steatosis, a result that closely resembles the results of a recently published study on the prevalence of Controlled Attenuation Parameter-detected steatosis in a population of patients with T2DM, reporting an overall prevalence of 72.6% [27]. While the link between obesity, T2DM, and MASLD is well established, we found a discordantly high prevalence of steatosis, highly suggestive of the premorbid role of fatty liver in insulin resistance and metabolic dysregulation. In line with previous reports regarding socioeconomic status being a risk factor for metabolic dysfunction, a recently published study revealed a significantly higher rate of steatosis (53.9% vs. 46.9%, *p* = 0.08) and steatohepatitis (23.1% vs. 16.2%, *p* = 0.03) in patients with a higher social deprivation index (comprising variables such as poverty, employment, housing, ethnic minority status, and education) [28], further reinforcing this presumption.

The observed disproportionately high burden of metabolic disease contrasts with the relatively low prevalence of infectious disease. The overall prevalence of parasitic infections in the screened population was on par with the nationally reported prevalence amounting to 5.7% [29], while access and vaccination rates for COVID-19 were even higher than the national average [30]. Notably, a large proportion of the study population had access to primary healthcare services, yet the rates of routine visits and presentations and preventive interventions were relatively low.

The evolution of disease patterns in vulnerable communities has witnessed a significant shift from infectious to metabolic ailments, reflecting broader societal changes in lifestyle and healthcare. Historically, infectious diseases have posed substantial threats, driven by factors such as poor sanitation, inadequate access to clean water, and limited healthcare infrastructure. However, as these issues have gradually been ameliorated, a concomitant rise in metabolic disorders has emerged. Shifts in dietary habits, sedentary lifestyles, and the prevalence of processed foods have contributed to an alarming increase in conditions like obesity, diabetes, and cardiovascular diseases within vulnerable populations. Socioeconomic factors play a crucial role, as disadvantaged communities often have limited access to nutritious foods and encounter barriers to physical activity [6,7]. Moreover, cultural shifts towards convenience-driven diets have exacerbated this problem. The consequences of this shift are profound, placing an additional burden on already resource-strapped healthcare systems and exacerbating health disparities. Addressing the spectrum of disease in vulnerable communities necessitates a holistic approach that considers not only infectious threats but also the complex interplay of socio-economic, cultural, and lifestyle factors contributing to the rise in metabolic disorders.

The current study lacks the power to support a causative link between the various at-risk behaviors observed and their clinical consequences. While we reported several correlations, the main purpose of this study was epidemiological. We acknowledge that our study lacked a comparison group and that the sample may not be fully representative of rural Roma communities. Furthermore, the size of our target population was significantly lower than expected. As such, we refrain from making definitive statements about the burden of metabolic risk factors in comparison to the general population. However, these findings can pave the way to financing more in-depth research projects specifically targeted to these communities as well as provide sufficient grounds for targeted public health interventions.

The main limitations of this study are inherent to its design, constituting a multifaceted screening tool. While a wide array of issues were detected using this design, this study lacks an in-depth characterization of each problem, such as a further investigation into the nuances of metabolic dysfunction and the determination of the full extent of the prevalence of metabolic syndrome and an adequate causative link. Therefore, the presumptions related to the impact of socio-economic status, marginalization, social inequality, and at-risk behaviors are mere inferences rather than strong causation links.

## 5. Conclusions

In conclusion, our study highlights the pressing need for targeted interventions within vulnerable rural Roma communities. Socioeconomic disparities, limited access to healthcare, and cultural factors contribute to the heightened burden of conditions such as obesity and diabetes. Addressing these health disparities requires multifaceted strategies, including improving healthcare infrastructure, nutritional education, and community engagement. By recognizing and understanding the unique challenges faced by these communities, public health initiatives can be tailored to mitigate the impact of metabolic diseases, fostering a more equitable and healthier future for vulnerable rural Roma populations.

## Figures and Tables

**Table 1 healthcare-12-00816-t001:** Descriptive presentation of the demographic and metabolic features of the study participants.

Variables	Median (Interquartile Range)/*n*, (%)
Age (years)	51 (38.25–63)
Gender (M)	94 (27.5%)
Height (cm)	1.62 (1.57–1.67)
Weight (kg)	72 (62–85)
BMI	27.34 (23.88–31.81)
Smoking (*n*,%)	159 (46.4%)
Alcohol (*n*,%)	38 (11.1%)
Sweet beverages (*n*,%)	175 (51.2%)
Obesity (No%)	113 (32.9%)
Degree of obesity (overweight, grades I, II, III (*n*,%)	122/61/31/19 (35.5%/17.7%/9%/5.5%)
HTN (*n*,%)	185 (54.1%)
De novo HTN (*n*,%)	66 (19.2%)
Random blood sugar (*n*,%)	108.5 (92–138)
DM (*n*,%)	99 (28.9%)
De novo DM (*n*,%)	67 (19.5%)
Liver steatosis (*n*,%)	111 (57.2%)
GP (*n*,%)	324 (94.5%)
Stable work place (*n*,%)	33 (9.6%)

Abbreviation: BMI, body mass index; DM, diabetes mellitus; GP, general practitioner; HTN, hypertension; SPB = systolic blood pressure; DPB = diastolic blood pressure.

**Table 2 healthcare-12-00816-t002:** Association between demographic and metabolic features and liver steatosis for this study group.

Variables	No Liver Steatosis (*n* = 83)	Liver Steatosis (*n* = 111)	*p*
Age (yrs)	47 (34.25–56.00)	60 (51.00–66.00)	<0.001
Gender (M)	66 (45.8%)	78 (54.2%)	0.1461
Alcohol (*n*,%)	7 (33.3%)	14 (66.7%)	0.3552
Sweet beverages (*n*,%)	46 (47.6%)	51 (52.6%)	0.2139
BMI (kg/mp)	25.39 (22.71–28.89)	30.61 (27.34–35.16)	<0.001
Obesity (*n*,%)	16 (21.3%)	59 (78.7%)	<0.001
Degree of obesity (*n*,%)	30/11/3/2 (43.5%/27.5%/14.3%15.4%)	39/29/18/11 (56.5%/72.5%/85.7%/84.6%)	0.0267
Random blood sugar (*n*,%)	101 (87.00–124.25)	122 (96.50–156.75)	<0.001
DM (overweight, grade I/II/III, *n*,%)	16 (25.4%)	47 (74.6%)	0.0007
HTN (*n*,%)	34 (30.1%)	79 (69.9%)	<0.0001
SBP (mmHg)	130 (119.25–143.00)	144 (130.50–158.00)	<0.001
DBP (mmHg)	84 (76.00–90.00)	90 (80.00–98.75)	<0.001
Stable work place (*n*,%)	8 (44.4%)	10 (55.6%)	0.8814
GP (*n*,%)	80 (42.1%)	110 (57.9%)	0.1893

Abbreviation: BMI = body mass index; DM = diabetes mellitus; GP = general practitioner; HTN = hypertension; SBP = systolic blood pressure; DBP = diastolic blood pressure.

## Data Availability

The original contributions presented in this study are included in the article/Appendix A; further inquiries can be directed to the corresponding authors.

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
