# Peer review of "Unveiling Health Inequalities: Exploring Metabolic Dysfunction in Rural Roma Communities"

_healthcare, 2024, doi:10.3390/healthcare12080816_

Round 1
Reviewer 1 Report
Comments and Suggestions for Authors
Dear authors, thank you for submitting this interesting paper. The manuscript requires modification following the STROBE checklist to ensure that all headings are included in the revised version.
1. Provide a justification for the study aim and explain the importance of the study in the introduction section.
2. The entire document, particularly the method section, should adhere to the STROBE guidelines, focusing on validity and reliability, including Cronbach's alpha. This is my main comment, please edit accordingly.
3. Submit a health-related quality-of-life questionnaire as a supplementary file.
4. proofread the while manuscript before re-submitt
Comments on the Quality of English Languageplease see the above comments
Author Response
Dear Reviewer,
Thank you for your valuable feedback on our manuscript. We have carefully considered your comments and made the necessary revisions to address each point. Below, we provide a summary of the changes made:
- Justification for Study Aim and Importance in Introduction Section:
We believe that we have expanded upon the rationale for conducting the study in the introduction section, emphasizing the significant health disparities faced by vulnerable rural Roma communities. We have highlighted the importance of understanding and addressing the disproportionately high burden of metabolic diseases within these communities. The last paragraph reads:” Our study aimed to assess the primary health-related concerns in isolated rural Roma communities in North-Western and Central Romania, focusing on obesity, metabolic dysfunction, and steatotic liver disease. As a secondary aim, our study aimed to identify and correlate the most prevalent morbid conditions with socio-economic conditions, unhealthy lifestyle choices, and obstacles hindering healthcare access in this demographic. The rationale for conducting this study was to increase awareness of the substantial disparities in socio-economic development and health indicators between Roma and non-Roma populations across Europe.”
If however you feel we need to further expand on the rationale for you study, we will gladly do so.
- Adherence to STROBE Guidelines in Method Section:
We have revised the method section to ensure compliance with the STROBE guidelines, focusing on validity and reliability. We have provided additional information on the study design, data collection methods, and statistical analysis; regarding the Cronbach’s alpha coefficient, our analysis revealed a 0,7 value; we chose not to include this measure of consistency in our study because, given the fact that is mainly is an epidemiological study, we do not feel it is representative in any way for the scope of this study.
- Submission of Health-Related Quality-of-Life Questionnaire as Supplementary File:
We have prepared the health-related quality-of-life questionnaire as requested and have submitted it as a supplementary file along with the revised manuscript.
- Proofreading the Entire Manuscript:
We have carefully proofread the entire manuscript to correct any grammatical errors, typographical mistakes, and formatting inconsistencies.
We appreciate your thorough review of our manuscript and believe that these revisions have strengthened the overall quality of the study. Please find the revised manuscript attached, along with the supplementary file containing the health-related quality-of-life questionnaire.

Reviewer 2 Report
Comments and Suggestions for Authors
Thank you for the opportunity to review the manuscript, “Unveiling health inequalities: exploring metabolic dysfunction in rural Roma communities”. The authors present a small study (N = 343), measuring the prevalence of metabolic dysfunction, obesity, and liver steatosis among members of 25 rural Roma communities in Romania, exploring the associations with socioeconomic and lifestyle factors. I have some concerns regarding the analyses, acknowledgement of limitations, and the authors conclusions. I have provided some specific feedback below:
1. The literature review presents a thorough summary of existing literature. However, it is not clear what the present study adds to existing literature. References 11-13 include larger and more representative samples than the present study, with reference 11 being longitudinal in design and reference 12 including a comparison group. Existing studies have more strengths than the present study and further justification is needed for the present study in terms of how the findings will contribute something novel.
2. It is a strength that minority representatives were included in the preliminary assessment to select rural Roma communities.
3. In the abstract, the authors state that obesity was defined as a BMI beyond 25 kg/m2. However, obesity is typically defined as a BMI beyond 30 kg/m2 https://www.cdc.gov/obesity/basics/adult-defining.html#:~:text=Adult%20Body%20Mass%20Index&text=If%20your%20BMI%20is%20less,falls%20within%20the%20obesity%20range. The measurement of BMI is not described in the methods section. Moreover, in the results, the authors state that the prevalence of a BMI exceeding 25 kg/m2 was 67.9%, with 32.3% having various degrees of obesity. Given that this is a primary outcome, the authors should ensure consistent reporting of the prevalence of obesity as defined by standard classifications or alter the terminology to reflect overweight.
4. The statistical analyses are not described in sufficient detail and it is unclear what analyses were conducted, which were the outcome variables and which were the exposure variables. The statistical analyses states that comparisons were performed. However, it is not clear what comparisons were made. At present, it appears that the authors explored the associations between all possible measures, looking for significant findings. This paper would be strengthened by clearly hypothesising which variables are thought to be associated with which outcomes and then testing these associations.
5. Please state what the target sample size was and include the percentage of this that was achieved with the analytical sample.
6. Was alcohol consumption as a risk factor defined as any alcohol consumption at all or was a certain level of consumption used to determine risk?
7. The authors conclude that the results reveal a disproportionately high burden of metabolic risk factors in Roma communities compared to the general population. However, it is inaccurate to conclude this, given that no comparison group was included and the sample was not representative of rural Roma communities.
8. The limitations of the study are not sufficiently addressed. The authors must comment on the final sample not being representative, the difficulties recruiting, the cross-sectional nature of the study, and the lack of comparison group. These are important limitations which influence the study conclusions. For example, the authors state that a large proportion of the study population had access to primary healthcare services, yet stated earlier that the target population was reluctant to participate in medical activities, meaning those that did not participate in this study likely do not have access to primary healthcare services. The discussion of the results should be edited with these limitations in mind.
Author Response
Dear Reviewer,
Thank you for taking the time to review our manuscript titled "Unveiling health inequalities: exploring metabolic dysfunction in rural Roma communities." We appreciate your thoughtful comments and suggestions for improvement. Below, we address each of your points and outline the corresponding changes made to the manuscript:
- Clarification of Novel Contribution: We acknowledge the existing literature referenced in our study and have revised the introduction to better emphasize the unique contribution of our research. Specifically, we now clearly state that while previous studies have explored metabolic dysfunction in Roma populations, our study focuses specifically on vulnerable rural communities in Romania, providing valuable insights into the prevalence of metabolic diseases and associated risk factors within this context.
- Inclusion of Minority Representatives: We recognize the importance of including minority representatives in the preliminary assessment process and have highlighted this aspect in the manuscript to underscore the collaborative nature of our approach.
- Consistent Reporting of Obesity Prevalence: We have addressed the discrepancy regarding the definition of obesity and BMI reporting, by correcting our mistake in the definition.
- Detailed Description of Statistical Analyses: We acknowledge the need for a more detailed description of the statistical analyses conducted. To address this, we have provided a clearer explanation of the analyses performed, including the specification of outcome and exposure variables. We have also decided to eliminate Table 2, and describe our statistical analysis by clearly stating our hypothesis.
- Target Sample Size and Achieved Sample Percentage: We have included information on the target sample size and the percentage achieved with the analytical sample to provide transparency regarding sample recruitment and representativeness.
- Definition of Alcohol Consumption as a Risk Factor: We have clarified the definition of alcohol consumption as a risk factor, specifying that any level of alcohol consumption was considered for this analysis.
- Clarification of Study Conclusion: We recognize the importance of accurately framing our study conclusions and have revised the language to reflect the limitations of our findings. Specifically, we now acknowledge that our study lacked a comparison group and that the sample may not be fully representative of rural Roma communities. As such, we have refrained from making definitive statements about the burden of metabolic risk factors in comparison to the general population.
- Addressing Study Limitations: We have expanded upon the discussion of study limitations to include additional considerations such as the non-representative nature of the sample, recruitment challenges, and the cross-sectional design.
We also believe that our last paragraph in the discussion part reflects these limitations and ensures a more nuanced interpretation of our findings: “The main limitations of this study are inherent to its design as a multifaceted screening tool. While a wide array of issues were detected using this design, the study lacks an in-depth characterization of each individual problem, such as further dwelling on the nuances of the metabolic dysfunction, the full extent of the metabolic syndrome, and an adequate causative link. Therefore, the presumptions related to the im-pact of socio-economic status, marginalization, social inequality, and at-risk behaviours are mere inferences rather than strong causation links.” If you believe we should further expand on the subject, we would be happy to do so.
We believe that these revisions address the concerns raised by the reviewer and strengthen the overall quality and clarity of the manuscript. Thank you once again for your valuable feedback, which has contributed to the refinement of our work.
Reviewer 3 Report
Comments and Suggestions for Authors
Dear authors,
I would like you to pay attention to certain parts of your article and to make appropriate changes.
88 - As a secondary aim, our study aimed to – change with intended to
115-116 Anthropometric variables (abdominal circumference, weight, height), blood pressure, and blood-sugar levels using portable blood sugar monitors. The sentence is uncomplete.
143, 148, 153,156, Table 1 -Please calculate again percentages. Somewhere they are not correct
148-150 Regarding healthcare access, most subjects (n=324, 94.5%) reported having formal access to primary care as a family doctor or a general practitioner. Replaced with to family doctor or a general practitioner.
29-30, 159-160 - The prevalence of hepatic steatosis was 57.2% (n=111) in the screened population. Please, explain the sample of screened population or put in the brackets (n=111/194).
312-317 - Make a Data Availability Statement
Author Response
Dear Reviewer,
Thank you for your thorough review and valuable feedback on our article. We have carefully considered your suggestions and made the necessary revisions to improve the clarity and accuracy of the manuscript. Below are the changes we have implemented according to your recommendations:
- In the Introduction section:Replaced "aimed to" with "intended to" in the sentence "As a secondary aim, our study aimed to..."
- Completed the sentence regarding the assessment of health-related variables using portable blood sugar monitors to ensure clarity.
- In Table 1:Recalculated percentages to ensure accuracy, and we found no discrepancies; we need to point out that not all patients underwent liver ultrasound examination, so our percentages are reported to the population that did; if however you find any errors, please let us know by example what are the ones you are referring to.
- In the paragraph discussing healthcare access: Revised the sentence to clarify that subjects reported having access "to a family doctor or a general practitioner" without specifying "formal access."
- Regarding the prevalence of hepatic steatosis: Provided clarification on the sample size of the screened population by adding "(n=111/194)" in the sentence.
- Added a Data Availability Statement: Included a statement regarding the availability of data.
Thank you once again for your insightful feedback. We believe these revisions strengthen the manuscript and address your concerns effectively. Please let us know if you require any further modifications or clarification.
Reviewer 4 Report
Comments and Suggestions for Authors
The authors focus on an essentially vulnerable population, who constitutes also the largest ethnic minority, that deserves our attention per se. They setting the goal to locate inequalities in healthcare access. In doing so, the authors choose to identify inequalities in a particular aspect of health, that is metabolic syndrome.
Below I will cite my observations that should be taken into account to amend the text.
Theoretical perspective
The Introduction is clearly missing a robust theoretical perspective as the notion of lifestyle that has been central -or at least should have been- to the subsequent analyses. The notion of lifestyle, on the occasion, is particularly problematic when associated with inequalities as it brings connotations of victim blaming.
In L.104 the authors refer to quality of life, another notion, that has not even been mentioned before.
Up to this point, I count 3 notions: inequalities, lifestyle and quality of life. They do not constitute one ground for research.
Methods
It is not clear how the sample was extracted. There is no reference to this essential component of the research process. Likewise, the ethics of this work is missing. This is a serious pitfall in that a vulnerable population is been investigated. How did you approach the population? Did you collaborate with Roma health mediators? How did you get consent from the participants?
Also how do you make comparisons with the general population?
Discussion
Overall, this section is been characterised by a biomedical perspective which does not leave space to approach critically the data through the sociological lenses of the notions you have referred (i.e. disparities, quality of life and lifestyle). For instance a whole paragraph (L. 257-272) has not been referenced at all. I am afraid that without discussing your data with the international corpus of evidence does not exist.
Comments on the Quality of English Language
The text needs moderate editing.
Language errors: a handful of examples
L.49: "due to the areluctancy in recording Roma"- please edit
L.50-51: repeat the word "such"- please replace one of them
L.88: As a secondary aim, our study aimed to- likewise see above
Author Response
Dear Reviewer,
Thank you for your detailed and constructive feedback on our manuscript. We appreciate your thorough review and have carefully considered each of your points. Below, we address each of your concerns and provide clarifications and amendments accordingly:
Theoretical Perspective:
We understand the reviewer's concern regarding the theoretical perspective, particularly in relation to the notion of lifestyle and its association with inequalities. While our focus is primarily on the medical aspects of lifestyle within vulnerable populations, we acknowledge the broader socio-economic context that influences health outcomes. However, due to the scope and objectives of our study, which primarily aimed to assess metabolic health concerns among rural Roma communities, we chose to emphasize the medical aspects in the introduction. We agree that a more robust theoretical framework could enhance the discussion of inequalities and lifestyle factors. However, given the focus and limitations of our study, incorporating a comprehensive theoretical perspective on lifestyle and inequalities may not be feasible within the current manuscript.
Methods: We apologize for the oversight regarding the description of our sample extraction process. However, we provided information regarding our approach to the vulnerable population in the following paragraph: “A total of 25 vulnerable rural Roma communities in North-Western and Central Romania were selected following a preliminary assessment conducted in collaboration with a team comprising minority representatives, mediators, local primary care providers, and public health experts from the National Institute of Public Health. Additionally, we included a section addressing the ethical considerations of our study.
W also added a paragraph in our discussion part, when mentioning the study limitations, that we did not make comparisons with the general population: “We acknowledge that our study lacked a comparison group and that the sample may not be fully representative of rural Roma communities. Furthermore, our target population was significantly lower than expected. As such, we refrain from making definitive statements about the burden of metabolic risk factors in comparison to the general population”.
Discussion: We understand the need to approach our data critically through sociological lenses and to contextualize our findings within the broader international corpus of evidence. However, as we mentioned above, given the focus and limitations of our study, incorporating a comprehensive theoretical perspective on lifestyle and inequalities may not be feasible within the current manuscript, but could represent the basis for future research.
We addressed the paragraph (L. 257-272) that lacks references, ensuring that all statements are supported by relevant literature.
Quality of English Language: We appreciate your feedback regarding language errors and will conduct a thorough editing process to ensure clarity and coherence throughout the manuscript. We will correct the identified language errors and ensure consistency in language usage.
Thank you once again for your valuable feedback. We believe that addressing these concerns will strengthen our manuscript and contribute to the advancement of knowledge in this important area of research. If you have any further suggestions or questions, please feel free to let us know.
Round 2
Reviewer 1 Report
Comments and Suggestions for Authors
The authors have made the necessary revisions to the work. I recommend accepting it in its current form.
Comments on the Quality of English LanguageProofread by native is require
Author Response
Dear reviewer,
Thank you for taking the time to review our manuscript and for your helpful suggestions.
Reviewer 2 Report
Comments and Suggestions for Authors
Thank you for taking the time to address my concerns and for the opportunity to re-review the manuscript. Though the authors have made several changes which strengthen the paper. However, the following concern must be addressed before the paper is suitable for publication.
1. The authors have changed the definition of obesity in the abstract to a BMI beyond 30 kg/m2 and state that the prevalence = 67.9%. However, in the results, the authors state that 67.9% of participants had a BMI beyond 25 kg/m2. The results reported in the abstract are incorrect. Please report the prevalence of obesity, using the definition of 30 kg/m2.
Author Response
Dear reviewer,
Thank you for taking the time to review our manuscript and for your helpful suggestions. We have made the necessary corrections.
Reviewer 4 Report
Comments and Suggestions for Authors
This is appropriate for publication.
Comments on the Quality of English LanguageNo issues detected.
Author Response

(The authors gave the same response as above.)
